# Functional Polymers Structures for (Bio)Sensing Application—A Review

**DOI:** 10.3390/polym12051154

**Published:** 2020-05-18

**Authors:** Kamila Spychalska, Dorota Zając, Sylwia Baluta, Kinga Halicka, Joanna Cabaj

**Affiliations:** Faculty of Chemistry, Wrocław University of Science and Technology, 50-137 Wrocław, Poland; kamila.spychalska@pwr.edu.pl (K.S.); dorota.zajac@pwr.edu.pl (D.Z.); sylwia.baluta@pwr.edu.pl (S.B.); kinga.halicka@pwr.edu.pl (K.H.)

**Keywords:** conducting polymers, molecularly imprinted polymers, composites, electrochemical sensing, microfluidics, lab-on-a-chip

## Abstract

In this review we present polymeric materials for (bio)sensor technology development. We focused on conductive polymers (conjugated microporous polymer, polymer gels), composites, molecularly imprinted polymers and their influence on the design and fabrication of bio(sensors), which in the future could act as lab-on-a-chip (LOC) devices. LOC instruments enable us to perform a wide range of analysis away from the stationary laboratory. Characterized polymeric species represent promising candidates in biosensor or sensor technology for LOC development, not only for manufacturing these devices, but also as a surface for biologically active materials’ immobilization. The presence of biological compounds can improve the sensitivity and selectivity of analytical tools, which in the case of medical diagnostics is extremely important. The described materials are biocompatible, cost-effective, flexible and are an excellent platform for the anchoring of specific compounds.

## 1. Introduction

An increasing interest in the development and operation of analytical devices for the detection, quantification, and monitoring of specific analytes has led to the appearance of sensors and biosensors. These devices are successfully gaining a growing number of recipients in the field of medical diagnostics, environmental protection and food safety, thanks to their high sensitivity, specificity and ability to analyze in real-time.

Every (bio)sensor is composed of two intimately associated elements: a receptor layer and a transducer. The receptor layer is built of (bio)recognition material, which recognizes the desired analyte, whereas the transducer converts the (bio)chemical signal resulting from the interaction of the analyte with the receptor into a digital electronic signal [1]. (Bio)sensors can be divided, for instance due to the transducer used (e.g., electrochemical or optical (bio)sensors). The electrochemical approach based on the voltammetric transducer is characterized by the ability to measure the current produced during the oxidation (or reduction) of a product (or substrate) when an increasing (or decreasing) potential is applied. The working of an optical sensor is based on measurements of quantitative changes of parameters characterizing light radiation (e.g., fluorescence). Electrochemical (bio)systems are nevertheless not as sensitive and selective in contrast to optical systems. Besides better results, which are obtained in the case of optical devices, they also possess relatively small size, which allows to construct compact devices. These tools are resistant to radio interference and other electromagnetic waves in comparison with electrochemical systems. It is worth mentioning that they usually show high sensitivity while simultaneously measuring without causing contamination with the reaction product [2]. The researchers have already presented a lot of fluorescence or electrochemical-based (bio)sensors that were successfully applied for heavy metal ions determination [3,4,5].

Very often (bio)sensor devices are fabricated in the form of micro-total analysis system (µTAS) or lab-on-a-chip (LOC). For fabrication of miniaturized LOC devices based on microfluidic and electrochemical sensing, a wide range of materials are used, such as glass, silicon, plastic, paper, hydro-gels, polymers and composites [6]. However, due to the low cost, flexibility and biocompatibility, polymeric materials represent a strong group. Some fabrication methods have been introduced for polymer-based microfluidic approaches, for example laser ablation, hot embossing or polymer casting [7]. Cyclic olefin copolymer (COC), polycarbonate (PC), polyvinyl chloride (PVC), poly(methyl methacrylate) (PMMA) and polydimethylsiloxane (PDMS) have been widely reported as substrates for construction of micro-devices for LOC applications [8].

Conductive polymers (CPs) have met with great interest since the properties of conjugated polyacetylene polymer were defined in 1977 [9]. Due to their diversity resulting from their chemical and physical properties, they can be used in various fields. Over the past two decades, polymeric materials that can be physically or chemically modified, e.g., by changing pH, introducing appropriate functional groups or modifying biologically active molecules, have been of great interest to scientists. Recent progresses in the controlled synthesis of these polymeric materials allow the rational design and preparation of new chemical and biological sensor systems. Such compounds are very often used, not only for manufacturing the (bio)sensor tools or LOC, but also because of their chemical and physical modification, thanks to which polymer-based matrices are capable of binding biologically active molecules, which is a crucial step in biosensor construction.

Currently, polymers can be in the form of solutions, gels, self-assembled nanoparticles, films or solids, so they can be used in controlled drug and gene delivery, catalysis, detection and imaging and self-repairing materials. Intelligent materials based on polymers, in contrast to their low molecular weight, have a number of advantages in terms of structural stability, dispersion in water, biocompatibility, ease of processing and integration with detection devices [10]. The dynamic, chemical, electrical and physical nature of conductive polymers can significantly improve the parameters of the sensors and biosensors [11]. Compounds such as poly (3,4-ethylenedioxythiophene) polystyrene sulfonate (PEDOT: PSS), polypyrrole (PPy) and polyaniline (PANI), due to their conductive properties, are widely used in the electrochemical (bio)sensor devices. Despite the fact that the conductive properties of polymers are crucial in the development of (bio)sensors, a large number of non-conductive polymers are also used as carrier matrices for the immobilization of biomolecules. Their advantages include excellent biocompatibility, low toxicity, and selectivity, all of which improve biosensor performance [12].

In this review, we present a wide range of polymer-based functional materials for (bio)sensors application, which are a part of LOC devices.

## 2. Idea of Detection Mechanism

Biosensor design, for LOC applications, combines various aspects, such as fabrication, immobilization, or transducing. Together, it offers multi-disciplinary research in chemistry and biology, ensuring extremely specific and independent of physical limitations (such as pH or temperature) devices [13]. For the development of effective biosensors, it is essential to select a suitable immobilization method that will ensure appropriate performances of a biosensor, such as good operational and storage stability, high sensitivity, high selectivity, short response time and high reproducibility. The crucial problem in the design of enzymatic electrodes is to enhance speed and reversibility of charge transfer between the enzyme and the electrode. Fortunately, the charge transfer in biosensors may be optimized with some mediating particles, such as using conductive materials for electrode modification (e.g., polymers). Such compounds improve the electron transport between the active site of the enzyme and the electrode surface, hence the sensor has short response time and high sensitivity. While transferring the electrical charge, polymeric materials suit as an adequate microenvironment for the anchoring of protein and as a transducer (Figure 1) [14,15,16].

In this review, we present a broad range of commonly used polymeric materials for sensing platforms development. CPs, composites and molecularly imprinted polymers (MIPs) are the most widely used materials for various sensing approaches, they are also applicable as a matrix for further sensing surface modification. They possess a possibility to work as an integrated part in one chip devices and biodevices, such as LOC, which are also described in this review.

## 3. Conductive Polymers

CPs, due to delocalized π electrons within the molecule, have unique properties, including electrical conductivity, low ionization potential, low optical transition energy, absorbance and light emission [17,18]. For this reason, they are used, among others in OLED diodes, solar cells or sensors [19,20,21,22,23]. The key issue in the construction of these devices is the use of materials with appropriate electrically conductive properties, which can be provided, for example by proper distribution of HOMO and LUMO orbitals. In addition, CPs due to their light weight, tunable conductivity, low cost and easy fabrication are an excellent solution for the production of the multifunctional sensors [24,25]. Various synthesis methods allow for rapid CPs development. Simple CPs structure manipulation, by changing various building blocks from simple phenyl units to heterocyclic aromatic units, as well as extended arenes, allows structures to be adapted to target properties, especially in sensors.

### 3.1. The Thin CP Films

In general, subtle structural differences between nonspecific CPs-based sensing elements allow for differential interactions with analytes that establish unique and identifying response patterns. These structural changes create a specific fingerprint that can be used to distinguish similar compounds using linear discriminant analysis (LDA) and major component analysis (PCA). These algorithms highlight and summarize the distinguishing features in large data sets, thus leading to the differentiation of chemical substances [26]. For example, CPs-based fluorescent sensors work through analytically induced energy transfer. Various conformational changes, aggregation processes are used to manipulate excitons, the scope of emission, as well as the process of quenching / enabling emissions [27]. They are distance dependent and require strong CPs-analyte interactions that are usually facilitated by the integration of molecular recognition elements (receptors) within or extended from the CPs skeleton. Still, research is ongoing on rapid, selective identification of the analyte. This requires spatially separate sensor units, each with its own recognition element, based on cheap and safe materials.

In 2016, Makelane et al. reported novel dendritic star-copolymer sensor system based on 3 poly(propylenethiophenoimine) (G3PPT) and poly(3-hexylthiophene) (P3TH) on a gold electrode (Au|G3PPT-co-P3HT) for the determination of phenanthrene (PHE) in oil-polluted wastewater (Figure 2) [28]. The G3PPT-co-P3HT-modified gold electrode was prepared by the electropolymerization of 3-hexylthiophene on a G3PPT-coated gold electrode using cyclic voltammetry (CV) for 8 polymerization cycles. The modified electrode was immersed in an electrochemical cell solution containing acetonitrile, 3-hexylthiophene (monomer) and 0.1 M Bu_4_NClO_4_. In the study they applied alternating current voltammetry (ACV) because of its good sensitivity and limit of detection (LOD) of up to 10^−7^ M when used in electroanalysis. As a result of the research, it was found that the sensor parameters include a dynamic linear range (DLR) of application values of 2.78–37.65 nM and a limit of detection as 1.42 nM (0.253 ppb). The WHO guideline value for polyaromatic hydrocarbons (PAHs) is 0.7 ppb, which falls within the DLR of the dendrimeric sensor for PHE.

Gurudatt and co-workers explored the preparation of a stable lipid-bonded polyterthiophene layer formed on the channel walls that provided a unique surface characteristic for the interaction with cancer cells (Figure 3) [29]. For this purpose, they used the electrochemical microfluidic channel (EMFC). The EMFC was prepared by screen printing the carbon ink on a glass slide. The amine functionalized conducting polymer precursor, [2,2′:5′,2″-terthiophene]-3′,4′-diamine (DAT) was electrochemically polymerized onto the channel walls to act as a substrate material to support the immobilization of the lipid molecules (phosphatidylserine). After separation, to detect the electrochemically inactive circulating tumor cells (CTCs), the tetracycline compound daunomycin (DM) has been selectively interacted with the cancer cell surface prior to the separation experiment. Different CTCs in blood were separated and detected amperometrically at the carbon electrode installed at the channel end, through mass and surface charge of target species, fluid flow rate, AC amplitude, and AC frequency. The separated CTCs were selectively detected via the oxidation of daunomycin adsorbed specifically at the cells, using an electrochemical sensor installed at the channel end. The authors demonstrated that the device successfully identified different cancers by the means of isolation and detection of CTCs in human blood samples. The lipid-modified conductive polymer on the microfluidic channel walls showed a remarkable increase in the specific interaction with DM-decorated cancer cells and was able to separate and detect the CTCs in a short time period of 400 s. Furthermore, two different cancer cells showed maximum separation in the channel. The device was able to separate a single cell with 92.0 ± 0.5 % efficiency. The results described in this study are a very promising method of cancer detection at an early stage.

More recently, Rahman and co-workers reported the synthesis of (*E*)-*N*′-(benzo[d]dioxol-5-ylmethylene)-4-methyl-benzenesulfonohydrazide (BDMMBSH) (Figure 4) [30]. These compounds were used to selectively detect heavy metal ions by electrochemical methods. The two synthesized ligands acted as conjugated molecules for the detection of Pb^2+^, where their nitrogen and oxygen atoms transfer electrons to Pb^2+^ because they have single electron pairs. Based on the electrochemical method, the current signals (CS) of the BDMMBSH modified on a GCE with 5% conducting Nafion polymer matrix sensor have changed significantly during Pb^2+^ adsorption. This new device may be a suitable analytical tool for designing sensitive and selective sensors for detection of toxic metals in the environment and healthcare.

### 3.2. Conjugated Microporous Polymer (CMP)

The three-dimensional polymer network backbone made of conjugated microporous polymer (CMP) exhibits excellent porosity, stable backbone structure and a number of other properties that show wide perspectives for use in many fields of science, such as optoelectronics [31], supercapacitors [32], and sensors [33,34]. However, the extremely poor solubility and processability of CMP, as well as severe fluorescence quenching due to aggregation, limit their practical applications. It has been reported that the performance of conjugated organic microporous materials can be improved by controlling their morphology [35]. CMPs are formed under kinetic control and are therefore amorphous and do not display long-range molecular order. Despite this lack of order, in 2007 J. X. Jiang and co-workers pointed out that it is possible to fine-tune the micropore size distribution and surface area by varying the length of the rigid organic connectors, as shown for ordered materials with the crystal structure (Figure 5) [36]. Their work began the research of CMPs because these polymers have good chemical and thermal stability in comparison with metal–organic and covalent organic frameworks (MOFs and COFs), and there is broad scope for robust anchoring of specific functionality to the networks through the carbon–carbon bond formation. The polymer networks are conjugated and there is a wealth of opportunity for producing microporous materials with useful coupled chemical, electrical, or optical properties.

An example of the use of CMP in sensors is the work of Wang and co-workers [37]. They reported the synthesis of two multifunctional conjugated microporous polymers (CMP-LS7–8) containing pyridine by Pd-catalyzed Suzuki coupling reactions (Figure 6). The two polymers showed outstanding porosity, *N*-donor sites of pyridine units, and extended π-conjugated structures. CMP-LS7–8 exhibited excellent performance on both the capture of volatile iodine and tetracycline sensing/removal from water with good reusability. The iodine uptake for CMP-LS8 is higher than most of CMPs reported to date. In addition, this study shows that CMP materials obtained by Langmuir methods can integrate two functions of sensing and removal of antibiotic tetracycline in one material.

The research group of Liu constructed a high-luminescence CMP film based on a novel dendrimer (TPETCz) featured by its central tetraphenylethylene core with aggregation-induced emission effect (Figure 7) [38]. High specific surface area CMP films were fabricated by a facile electropolymerization method. These CMP films exhibit sensitivity to volatile organic compounds (VOCs). The scientists controlled the morphology and thickness of CMPs by adjusting the electrochemistry parameter. Further, a fluorescence array sensor was constructed based on the different fluorescence responses of CMP films and spin-coated films to VOC vapors, and selective detection of 18 types of VOCs was achieved by LDA analysis. This is one of the reports on the specific detection of VOC vapors and it shows that the fluorescent arrays have great potential applications.

Zhang et al. recently described new three fluorescent porous organic polymers containing pendant *N*-benzylcarbazole, *N*-benzyldibromo-carbazole, and *N*-benzyldimethoxy-carbazole groups (PAN-C, PAN-C–Br, and PAN-C–OCH_3_) for detecting pesticides (Figure 8) [39]. The scientists have shown that the three polymers show many different fluorescent responses toward the pesticides including trifluralin, isopropalin, glyphosate, fenitrothion, imidacloprid, and cyfluothrin. The reason for this is attributed to the different porous/chemical structures of polymers and the molecular sizes of the pesticides. It is interesting to observe that the test paper prepared from the polymer displays the rapid fluorescent response for pesticides. Moreover, after twelve cycling uses, the sensitivity of the test paper is almost unchanged, showing the potential application in detecting the residual toxic pesticides.

### 3.3. Polymer Gels

The gel has a continuous structure that is stable on an analytical scale of time and is constant in its rheological behavior [40]. Gels consist of two phases: a liquid-like phase and a solid network resembling a solid which surrounds the liquid, thus preventing its flow [41]. The gel contains about 99% liquid and about 1% solid (called a “gel”). From a constitutional perspective, gels can be classified as polymer and molecular gels. In the case of polymer gels, gelators are long chain polymers that form the three-dimensional network required for gelation by covalent or non-covalent cross-linking. Depending on the origin, polymeric gelators are classified as natural, e.g., agarose, gelatin or synthetic e.g., poly(acrylic acid), poly(ethylene glycol), etc. [42]. Conductive gels have spectacular advantages, such as adhesion, porosity, swelling and good mechanical properties, compared to those of bulk conductive polymers. The porous structure of the gels allows for easy diffusion of ions and molecules, swelling provides an effective interface between molecular chains and solution phases. Due to these properties, conductive gels are promising materials for many applications, such as energy processing and storage, sensors, medical and biological devices, superhydrophobic coatings, etc. [43,44,45]. Gels can be classified as organogels and hydrogels, depending on the type of solvent trapped in the gel. Organic solvents are trapped in organogels, but in hydrogels, water remains trapped because it acts as a gelling agent. Organogels are very useful, especially for the production of porous materials. Recently, there has been a huge increase in the area of research on polymeric and peptide hydrogels in relation to their use in various fields, especially in the field of energy and biomedicine [46,47,48]. Mainly hydrogels with conductive polymer PANI, PPy, polythiophene (PTh), and PEDOT are widely studied for their conductivity in the area of semiconductors [49,50,51,52].

The Luo research group has prepared a hard and functional gel that can find potential applications in the repair of cartilage, artificial skin, and sensors [53]. They received hydrogels with polyvinyl alcohol (PVA) and a polymer extracted from grape seed (GSP). The hydrogel was first cross-linked through crystalline PVA regions upon freezing-thawing cycles, and then immersed in ammonium sulfate (AS) solution to induce the hydrophilic hydrophobic transition of GSP. As a result, ultra-strong (tensile stress = 20.5 MPa), anti-fatigue (>95%), self-healable (healed stress = 10.0 MPa) and conductive hydrogels were obtained. The strength and toughness of the gel obtained are currently among the highest in the literature. In addition, the swollen gel, even after soaking in deionized water for 36 h, continued to maintain tensile strength at the megapascal level, surpassing most water-rich hydrogels. Scientists have chosen the GSP polymer because it is a natural, biocompatible and multifunctional macromolecule, which contains carboxylic acid, an amine and phenolic hydroxyl groups in their side chains. These groups facilitate the formation of many hydrogen bonds with PVA, which affects the properties of the obtained hydrogel.

Another research group studied a pH-responsive, *N*-(2-aminoethyl)methacrylate (AEMA)-based hydrogel, that was fashioned into an impedimetric pH sensor for the continual measurement and monitoring of tissue acidosis that can arise due to hemorrhaging trauma [54]. They indicated that the AEMA hydrogel had the highest sensitivity containing the appropriate pathophysiological range (pH 7.0–8.0). It was found that 1 mol% AEMA is the most robust, sensitive and shows optimal sensitivity in the field of pathophysiological detection of pH (7.35–7.45) for hemorrhagic trauma. This composition was designed as a responsive membrane on a microlithographically produced interdigitated microsensor electrode, and sensitivity was determined by R(QR)(QR) analysis. Non-freezable bound water has been found to be the most strongly correlated factor regulating the pH response of hydrogels.

Ginting and co-workers developed self-healing PAA/PPy-Fe composite hydrogels with antimicrobial and conductive properties through one-step polymerization [55]. The application of PAA/PPy-Fe composite hydrogels was evaluated by constructing a simple electrical circuit consisting of PAA/PPy15-Fe connected with a light-emitting diode (LED), which was powered by 3 V batteries. Composite hydrogels PAA/PPy-Fe have satisfactory self-healing properties, bactericidal ability, and high conductivity. The resulting PAA/PPy-Fe composite hydrogels were tested for their antibacterial activity on *Escherichia coli*. It was found that at higher pyrrole concentration PAA/PPy-Fe composite hydrogels are bactericidal. In addition, due to the conductive properties of PAA/ PPy-Fe hydrogels, the hydrogel conducts electricity by lighting an LED in the electrical circuit.

## 4. Composites

One of the most commonly used polymers for microfluidics applications, including LOC devices, is PDMS. Its properties, e.g., optical, mechanical, chemical or toxic, show undeniable advantages for LOC design and fabrication [56]. Nevertheless, some disadvantages are also listed. The most important of these include lack of electrical conductivity, low thermal conductivity and the fact that in the fabrication process, adhesion between PDMS and metal is very weak [57,58]. To overcome these problems, composites can be used. Two components form polymer composites: a polymer that provides physicochemical properties and a (nano)material embedded in its structure [59]. Proper selection of new, additional materials can complement missing properties of polymers.

Chałupniak and Merkoçi created a multilayer lab-on-a-chip device for heavy-metals preconcentration and electrochemical detection, using graphene oxide (GO) and PDMS (GO-PDMS) [59]. GO allows the adsorption of heavy metal ions. The created device had three layers, i.e., screen-printed carbon electrode (deposition and detection of metal ions), PDMS channel (detection) and GO-PDMS channel (preconcentration). The latter is the starting point, where adsorption onto GO molecules takes place, then a buffer causes desorption of metals and their movement to detection chip. Preconcentration can improve the sensing, lowering limit of detection even 30 times. The authors used Pb (II) as an example of heavy metal ions and seawater as a real sample. The results showed adsorption of over 98% of added Pb (250 ppb) on the composite surface, and desorption of over 96% Pb. The square-wave anodic stripping voltammetry (SWASV) showed that the pollutions in the sample affecting measurements can be purified using the GO-PDMS platform. Other tests proved that when alone, both copper and lead were fully adsorbed, however when mixed, copper had lower affinity to graphite oxide surface.

Another material used for composite preparation is carbon black. Brun et al. used a moulding method to create and test carbon black-PDMS electrode for electrochemical sensing [60]. Experiments were conducted using a three electrode system (prepared electrode as working electrode, Ag/AgCl reference electrode and Pt wire as an auxiliary electrode) in Fe(CN)_6_^3−^/KCl solutions. Conductivity values of this electrode depended on carbon concentration: when the concentration was increased from 10% to 25%, conductivity changed from 8.4 × 10^−6^ to 10.3 S·m^−1^. A minimum of 18% carbon was necessary for the conductivity to be significant. CV showed that higher carbon content favours electrochemical experiments. Obtained electrodes also presented low capacitive current. In addition, the authors confirmed that oxygen plasma treatment, a commonly used technique, can be applied to seal microchannels on carbon black-PDMS.

Niu et al. introduced a method of patterning conductive structures using carbon black-PDMS composite (C-PDMS) [61]. They mixed carbon black particles with PDMS and put it in a mould, baked, cured and washed. Pure PDMS gel was then poured over prepared conductive composite. The conductivity of obtained material as a function of concentration was tested. A threshold concentration value was determined (10% wt), beyond which increase in conductivity was observed. Resistivity was also tested, and the results showed an increase of resistivity with increasing temperature. Performed experiments proved that this composite has good both electrical and mechanical properties.

In the same work, Niu et al. used a composite consisting of PDMS and silver particles (Ag-PDMS) [61]. Ag particles size was approximately 1–2 µm. The same experiments as in the case of carbon black composite were carried out. Conductivity tests determined threshold concentration value at 83%, conductivity was higher than in C-PDMS. Resistivity tests showed an increase with increasing temperature to 120 °C and a decrease beyond that point, in contrast to all-rise tendency in C-PDMS. The authors also created 3D microstructures with the PDMS-based composites, using a soft lithography technique. The functionality of created circuit was tested with light-emitting diodes, proving the electric connection to be stable. This gives potential applications of the composite in microfabricated devices.

Silver particles were also used to create a microchip that would allow temperature control. The described integrated microchip combined a giant electrorheological-fluid (GER) actuated micromixer and micropump with microheaters all formed via soft lithography. Microheaters were fabricated with silver particles-PDMS composite and combined to form a heater array. Silver concentration in the composite was 86.3%. The prepared substance was put into a mould and baked. Better conductivity was achieved by thermal treatment. The heater array was used as part of a device for DNA amplification. One of the most important elements in this process is the temperature control, which is why temperature sensors were also included in the chip. A polymerase chain reaction (PCR) was successfully conducted to test created device [62].

Another noble metal used in composites is gold. PDMS is chemically inert, therefore attaching biomolecules to its surface presents difficulties. To overcome them, composites that introduce new properties are used. Zhang at el. proposed a method of in-situ synthesis of PDMS-gold nanoparticles (PDMS-AuNP) films for application in microfluidic systems [63]. PDMS films were incubated with HAuCl_4_ solution to create a composite. The film was changing color to shades of red when AuNP were formed. Gold nanoparticles were also synthesized on the surface of PDMS chips using different techniques (microchannels and oxygen plasma). These methods allow to pattern NP on the PDMS surface. To test created films, the authors performed an antibody-antigen binding. They used mouse IgG to bind to AuNP and fluorescein isothiocyanate (FITC) conjugated mouse anti-IgG for visualization. Fluorescent images that proved the pattern were obtained. In another test, an enzyme—glucose oxidase—was immobilized on the microchannel. This allowed for amperometric glucose detection using composite-based device. The microchip consisted of a three electrode system that included carbon fiber, Ag/AgCl and Pt wire as working electrode, reference electrode and auxiliary electrode, respectively. Phosphate buffer was used as medium. Enzyme reaction with injected glucose solution created hydrogen peroxide, which was detected at the working electrode.

AuNPs were also used for LOC biosensing of growth hormones. The biosensor detected antigen-antibody interactions. The PDMS microfluidic chip was made using a mould (spin-coated, baked and silanized SU-8) To create the composite, PDMS samples were incubated with HAuCl_4_ solution, which was introduced into the microchannels of the chip as well. Gold NP were achieved because of curing agent in PDMS. To fabricate a biosensor, bovine somatotropin (bST or BGH—bovine growth hormone) antibodies from mouse were attached to AuNP through EDC/NHS reaction, often used for coupling. Bovine growth hormone was introduced as an antigen. Localized surface plasmon resonance (LSPR) spectra were measured at each step of the process to test biosensing abilities of the device. Immobilization of biomolecules on gold nanoparticles lead to increase of refractive index of the medium. This caused the LSPR band to shift to longer wave lengths with the experiment progress. When attaching the antigen, the shift was caused mostly by interaction between antigen and antibody. To determine limit of detection, different concentrations of BGH in PBS solution were examined. LOD was found at 3.7 ng/mL (185 pM), which is low enough to apply in industry, as the concentration of BHG in milk can be below 10 ng/mL. The authors suggested that this method could be applied for detection of different polypeptides and proteins [64].

A different research group employed PDMS-AuNPs composite for biosensing purposes as well. They fabricated microfluidic chip by sealing together two flat composite films and used it in an electrochemical immunosensor for simultaneous detection of dual cardiac markers: cardiac troponin I (cTnI) and C-reactive protein (CRP). Antibodies for both proteins were conjugated with CdTe (CdTe-anti-cTnI) and ZnSe (ZnSe-anti-CPR) water soluble quantum dots (QDs), which served as the media. PDMS-AuNP microchip was obtained by incubating cured PDMS with HAuCl_4_. Sandwich enzyme-linked immunosorbent assay (ELISA) was applied. Antibodies (Ab_1_) were adsorbed on the composite surface, which allowed for antigen-antibody interaction. QDs-conjugated antibodies (Ab_2_) could then attach to the antigen. The detection of antigens was based on metal ions dissolved from QDs—the solution containing ions was examined by square-wave anodic stripping voltammetry in a three-electrode system. The sample volume required to perform experiment was much lower than in conventional biosensors (nanolitre compared to microlitre). The limits of detection were determined at approximately 0.004 µg/L for cTnI and 0.22 µg/L for CRP. This system could potentially be applied in different fields requiring heavy metal detection, such as environmental monitoring [65].

Ozhikandathil et al. took a different approach and fabricated an optical lab-on-a-chip device for ammonia and amino acids detection using ninhydrin-PDMS composite. Ninhydrin reacts irreversibly with ammonia ion, resulting in a purple, stable product. In the case of the composite, a change of color from yellow to red-pink is visible after exposure to ammonium hydroxide solution. The composite film was created by spinning the PDMS-ninhydrin mixture on silicon wafer. Oxygen plasma bonding was used on a glass substrate to fabricate a platform, and plastic tubes served as the guiding assembly for the gas. To fabricate the optical sensor, apart from ninhydrin-PDMS, the light emitting diodes and photo resistors were also used. The conducted experiments showed that at some point the sensor gets saturated, which was visible as two regions on the response time graph. To optimize properties, films of different thickness were tested. Results indicated that reducing the thickness can shorten the response time. The limit of detection was determined at 2 ppm. The authors also investigated the detection of amino acids, using glycine as an example. The reaction of ninhydrin with glycine was slow, accelerated by heating; a slight change of film color to blue was noticed. A change of absorbance peak was also observed, depending on the substrate: 420–480 nm for ammonia and 570 nm for amino acid [66].

Although PDMS is the most common polymer used in composites for lab-on-a-chip applications, it is not the only one. For example, Xu et al. developed a solid-state electrochemiluminescence sensor, using a composite of graphene and Nafion functionalized with poly(sodium 4-styrenesulfonate). The measurements consisted of CV and electrochemical impendence spectroscopy (EIS). Tris(2,2′-bipyridyl)ruthenium(II) (Ru(bpy)_3_^2+^) was immobilized on the film, limit of detection for tripropylamine was determined at 5 nM [67]. Rattanarat et al. created a droplet-based microfluidic sensor using a carbon paste electrode modified with graphene-polyaniline composite. The electrochemical sensor worked in a three electrode system (carbon paste electrodes, in modified in the case of the working electrode) and the measurements included CV, square-wave voltammetry (SWV) and chronoamperometry. Sensor was used for 4-aminophenol detection in pharmaceuticals containing paracetamol. They obtained LOD at 15.68 µM [68]. Dou et al. developed DNA-modified graphene oxide nanosensor that was integrated with paper-poly(methyl methacrylate) microfluidic chip. The device was used for multiplex quantitative loop-mediated isothermal amplification (LAMP) detection, which is a method of amplification of nucleic acids. *Neisseria meningitidis* and *Streptococcus pneumoniae* were used as pathogen samples and detection limits were found at 6 copies and 12 copies per assay, respectively [69].

All the listed examples prove that introducing a new material to a polymer base, such as metal or carbon particles, thus creating a composite, can improve the properties of the material and allow previously unreachable modifications. This can lead to better, quicker and more selective LOC devices.

## 5. Molecularly Imprinted Polymers

Fast and sensitive detection of pathogens, biomarkers or toxins is extremely important in modern medical diagnostics, environmental protection, and the food industry. A particular challenge is when the concentration of the analyte is very low. Although conventional technologies such as PCR, chromatography, mass spectrometry or ELISA are able to accurately detect the target molecule, unfortunately, these tests are tedious, inefficient and expensive. They also require not only trained personnel but also expensive instruments that can only be used in laboratory conditions. As a result, there is a growing interest in the development of portable and economical sensors, characterized by high sensitivity, selectivity, and quick response. Recently, MIPs are in the spotlight due to their unique chemical and physical properties and easy modification.

### 5.1. Molecularly Imprinted Polymers (MIPs)

MIPs are a type of polymers that contain specific molecular recognition sites-they have cavities in their structure that mimic the function and morphology of antibodies and receptors, thanks to which they are able to recognize target analyte molecules. MIPs are obtained by two basic methods: electropolymerization or standard free radical polymerization. As a result of free radical polymerization, a rigid monolith is obtained, which is then ground and sieved to obtain particles. Free radical polymerization is a simple method to obtain a large-scale product. Despite these advantages, this method is not free from disadvantages, which include low process efficiency, slow mass transfer and heterogeneity of the resulting molecules [70,71]. Electrochemical polymerization has a number of advantages over traditional methods, such as speed and ease of synthesis, the ability to carry out reactions in an aqueous environment, the ability to control the thickness and morphology of the obtained layer, and better coverage of the transducer surface. Unfortunately, there are also factors that significantly hinder the use of electropolymerization for the synthesis of MIPs. They include problems in removing the matrix and simultaneously optimizing both the printing and rebinding of the molecules [72].

The MIP synthesis process can be presented in several stages (Figure 9). This process is based on the fact that single molecules (monomers) co-polymerize with cross-linking monomers in the presence of standard molecules (of the target substance to be detected) until their free functional groups are depleted [73]. Matrix-functional monomer complexes are formed by reversible covalent bonds [74] or non-covalent interactions such as ionic, dipole-dipole or hydrogen bonds [75]. The functional groups of the introduced reference molecules are immobilized in the polymer mass [73]. A rigid polymer network is created, which is responsible for maintaining the resulting spatial structure [76]. Next, the reference molecules are removed by washing them [74], which results in impressions on the polymer surface corresponding to their functional groups [73]. These impressions are binding sites complementary in terms of size, shape and chemical functionality to the reference substance used [76].

This process modifies the MIP surface and is responsible for its functionality. This creates a three-dimensional polymer matrix with high specificity and selectivity for a given substance and its structural analogs [73]. In addition, compared to biologically active elements, such as antibodies or biological receptors, MIPs show much better mechanical properties, higher chemical and physical stability, and high resistance to temperature changes. Their synthesis is relatively easy, and preparation time and cost of production are much lower compared to other commonly used methods [77].

### 5.2. Selection of the Appropriate Functional Monomer

An extremely important stage in the production of MIPs-based sensors is the selection of the appropriate functional monomer due to the need to create direct bonds between monomers and target analyte molecules constituting the matrix [78]. Matrix molecules bind to monomers via covalent or non-covalent bonds. After cleavage of the covalent bond between matrix molecules and specific groups within the monomer molecule, free covalent bonds can be rebound in the presence of the target molecule. This is a very stable binding that greatly increases the specificity of the sensor. However, the slow and insufficient dissociation of the covalent bond, as well as the rigid polymer network resulting from the presence of strong bonds, additionally hinders the binding of target molecules, which results in a negative impact on the detection process. Non-covalent bonds (such as hydrogen bonds, ionic bonds or van der Waals forces) become an alternative, the use of which allows easier binding and removal of the matrix from the polymer particles. Therefore, MIPs using non-covalent bonds are more common in the literature [79,80].

### 5.3. MIPs-Based Sensors

There are three basic types of MIP-based sensors (Figure 10):

Sensors with one type of functional monomer (usually MIP) are produced using one monomer by free radical polymerization. Although some monomers, such as methacrylic acid (MAA), acrylic acid (AA) or acrylamide (AA) are commonly used to produce other types of sensors using MIP, few of them are used to design sensors based on only one type of monomer. Among the above-mentioned, MAA is the most commonly used monomer for the production of this type of sensors, due to the ability to form hydrogen and ionic bonds. Sensors with one type of functional monomer are used for small matrix molecules [77,81].

Sensor with many types of functional monomers: in most cases MIP-based sensors are prepared using more than one functional monomer due to the possibility of generating sufficient places of interaction between monomers and matrix particles. This solution is used for macromolecules, such as proteins or peptides, which have many functional groups on the surface. The presence of these groups makes it possible to form hydrogen bonds, ionic bonds or van der Waals interactions. Very often one type of neutral monomer is used as the backbone stabilizing a matrix made of another hydrophobic polymer synthesized in the presence of the target analyte. The presence of such a spine significantly increases the affinity of the cavity for the target molecule and stabilizes the structure of the resulting MIP, thanks to which it is possible to obtain optimal sensitivity and selectivity of the constructed sensor [81].

Sensors with many types of functional monomers in a complex with biologically active molecules: scientists are increasingly attracted by the integration of functional biological molecules with MIPs to improve the performance of MIP-based polymer sensors. Although the target molecules and printed complementary cavities ensure the selectivity of the sensor, monomers in combination with biologically active molecules such as aptamers or antibodies are able to significantly improve the selectivity of the sensor and its binding ability to the target molecule [77].

### 5.4. Recent Advances in Molecularly Imprinted Polymer Based Sensors

The following table (Table 1) summarizes the latest publications on the detection of various analytes by MIP. The table contains information on the functional monomer used, the material of the electrode used, the detection method and the limit of detection.

Jafari et al. developed a selective and sensitive MIP-based method for determining the concentration of the azithromycin (AZT) antibiotic [83]. Firstly, GCE was modified with GO and gold nanoruchins (GNU). In order to form MIP-film onto the surface of the GCE/GO/GNU electrode, the electrode was immersed in the mixture containing HNO_3_, H_2_SO_4_, aniline, and AZT. The MIP film was electropolymerized using CV for five cycles. Finally, the AZT molecules could be extracted through the immersion of the electrode in an aqueous medium. The research was carried out using electrochemical methods such as field emission scanning electron microscopy (FESEM), CV or EIS. The sensor constructed in this way showed good selectivity and sensitivity, a wide linear range and LOD enabling antibiotic determination even at the level of 0.1 nM.

Lahcen et al. proposed a fast and sensitive electrochemical sensor for detecting *Bacillus cereus* spores [84]. The sensor fabrication was based on the electropolymerization of pyrrole in a solution of LiClo4 using CV for five scans. After that, the solution of bacterial spores was added to the reacting mixture and the CV was continued for the next five scans. In the last step, the spores were extracted from the electrode surface by sonication or incubation in a surfactant solution. All electrochemical measurements were conducted in liquid medium using CV, EIS, and differential-pulse-voltammetry (DPV) methods. Such constructed sensor showed an excellent affinity for *B. cereus* spores and enabled detection with LOD at 10^2^ CFU/mL.

Teng and the team also developed an electrochemical sensor based on MIP [85]. In the first step, GCE was modified with PPy in order to obtain PPyNWs/GCE electrode. Such prepared electrode was immersed in the solution of phosphate buffer, dopamine (DA) (acting as the matrix), and OPD, and using the CV method the thin film of MIP was created. The DA molecules were removed from the template by incubation of the electrode into ethanol. Under optimized conditions, the calibration graph showed linearity in the concentration range of 50 Nm–100μM, while the LOD was 33 nM. The sensor is characterized by high sensitivity and selectivity in the determination of dopamine in both laboratory and pharmaceutical samples.

Gholivad and the team designed a sensor for electrochemical determination of ganciclovir based on electropolymerized 2,2-dithioaniline on the glassy carbon electrode surface, previously modified with multifunctional nanotubes [88]. The process of preparation of the sensing platform started with the modification of GCE with multiwalled-carbon nanotubes (MWCNs). In the next step, such prepared electrode was immersed in the reacting mixture containing phosphate buffer solution, 2,2-dithioaniline as a monomer, HCLO_4_, and the GCV as the template. MIP-film was created using the CV method (10 scans). After the reaction, template molecules was removed using the same method in phosphate buffer until a stable cyclic voltammogram was obtained. The measurements were carried out using the methods of CV and differential-pulse anodic stripping (DPAS). After optimization, the system was characterized by high sensitivity and selectivity towards GCV, and therefore the sensor could be used for drug detection in real samples with a very good LOD (1.5 nM).

Liu et al. designed and synthesized the MIP/GO nanocomposite that was used for electrochemical detection of testosterone [96]. In order to increase the surface area of the interaction and the sensitivity of the system, the group used GO sheets. GCE, previously modified with GO, was immersed in the mixture of acetate buffer solution, OPD (acting as a monomer), and testosterone (acting as a template). The MIP film was created by electropolymerization using the CV method (30 cycles). The testosterone molecules were removed by immersing the electrode in the ethanol solution. The testosterone detection process was carried out using the EIS method, which, thanks to its high sensitivity to surface interactions, enabled testosterone determination in a very low concentration range with a LOD of 0.4 fM.

## 6. Application of Polymeric Materials in LOC Devices

Nowadays, a clear increase of interest and high progress in microfabrication of LOC technologies had been observed. LOC is currently the most absorbing technology idea due to development of constant, fast and sensitive devices, which provide monitoring of crucial parameters in many fields of industry. A main area of development for these approaches is bioanalytics and integrated concepts for medical diagnostics, especially for the control and evaluation, e.g., health status of patients.

Basically, LOC or micro total analysis systems (μTAS) are hybrid devices combining micro or nano-sized multiplexed channels and electronic components for performing fast and sensitive measurements [98]. Such devices are able to operate in ”multi-mode” system due to the presence of microchannels, which allow liquid samples to flow inside the chip, but they also integrate measuring, sensing and actuating elements such as microvalves, microfluidic mixers, microelectrodes, thermal elements, or optical instruments [99]. Evolution in LOC technology is connected with adoption of a lot of sensing methods in these devices, such as capillary electrophoresis [100,101,102], electrochromatography [103,104,105], electrochemical detection [106,107], ultra-sound waves [108], mass spectrometry [109], scattering [110], absorbance [111] or fluorescence [112]. In consequence of combining many different analytical techniques inside one chip, these instruments ensure wide range of sample diagnostics, e.g., handling, mixing, filtering, monitoring and detecting. Miniaturization and integration of important components results in many advantages: reduced time of analysis and laboratory processes, automatization of measurements, compactness, portability, embedded computing, automated sample handling, low electronic noise, limited power consumption and low samples contamination (closed devices) [113]. LOC also represent a very promising scientific method for the construction of new generation of wearable, portable and implantable bioelectronic devices for point-of-care (POC) testing application.

Advancement of LOC devices can also be connected with the design and construction of platforms which will be able to selectively detect specific analytes. Chemical and biological sensors were originally studied at the microscale as an expansion of LOC/microchip engineering [114]. Many described biosensing pathways possess the ability of miniaturization to use in LOC devices. For specific recognition of specific samples, very often biologically active compounds, such as enzymes [115,116], whole cells [117,118] or antibodies [119] are employed. However, to ensure stable and effective anchoring of biomolecules, polymers are frequently used (Figure 11). They possess an evident advantages in comparison with other species, due to flexibility in design and cost effectiveness [120].

All designed instruments for LOC application should be able to be easily maintained, which is connected with convenient and simple channel washing and sample filling. All chip elements should be able to undergo easy manipulation, calibration, assembly and utilization by the laboratory personnel. The fabrication of these devices is comparable to the manufacturing of microelectronics chips. It applies photolithography technique, where patterns that represent microfluidic and electric features are moved onto a substrate by ultraviolet illumination via a chromium photomask [113].

In the current literature most of scientific reports concern electrochemical microsensors with microfluidics [121,122,123,124]. Manipulation of small volumes of fluids using channels in nano- or pico-scale in connection with simple and known electrochemical sensing is very attractive and is becoming the strongest profit of these kind of LOC devices [125]. In the case of electrochemical sensing, plug-based microfluidics are more advantageous than continuous flows of liquids. In such systems the volume of each plug can be highly reduced, multiple plugs can be processed in a single flow channel and the mixing of ingredients may be very rapid (<1 s) [126]. Very often plugs of aqueous solutions were created by injecting one solution into another immiscible carrier liquid using a flow channel [127]. Due to this, an electrode surface in the electrochemical measuring system may be contaminated by the carrier liquid when using this method. This problem of contamination can be mitigated by the separation of the plugs by air [126,128].

Electrochemical devices, along with many advantages, also ensure a safe and easy technique of placing biologically active material directly on the chip. This procedure opens new possibilities for the selective detection of a wide range of different analytes and presents a new area of LOC application. However, the biggest challenge for the design and construction of bioplatforms for LOC devices is the immobilization of the biological material. Such anchoring requires a suitable matrix, which should not only effectively bind the biomolecule on its surface, but also prevent the weakening of its catalytic activity and ensure stability in sealed microenvironment. Polymers-based materials represent a strong group of materials in this context, as described in this review. In this paper, we present a few designed polymer-based bioplatforms, which, when connected with electrochemical sensing pathways, create a new LOC device.

In electrochemical sensing approaches, different detection techniques can be employed, for instance, potentiometric and impedimetric methods. However, in potentiometric devices, very often ion-sensitive electrodes are used, which in most cases allow for pH changes measurements, and do not require any additional materials. Due to this, in this review we describe LOC devices based on the impedimetric method, where polymers and biological receptors are present. Manczak et al. reported an impedimetric microfluidic device for sensitive immune-detection of cells [129]. The sensitivity of the described method was estimated as the ability of the electrodes to trap monocytes by specific immune-reaction with CD14 antibody immobilized on the modified-electrode surface, resulting in a change in impedance The electrode had been modified with mixture of 11-mercaptoundecanoic acid (MUA), mercaptohexanol (MH) and a mixture of aqueous solutions of *N*-hydroxysuccinimide (NHS) and 1-ethyl-3-(3-dimethylaminopropyl-carbodiimide) (EDC). On the electrode prepared this way, an antibody has been immobilized. This proposed method allowed for the obtaining of the detection limit of 5 cells/mL, and thus presents a very promising sensing approach for medical diagnostics.

An et al. described the detection method for monitoring of CTCs, which allow for the early diagnosis of cancer [130]. In this investigation, they used a benzoboric acid modified gold-plated polymeric substrate (based on MUA and 3-aminophenylboronic acid (MBA)) with a regular 3D surface array (Figure 12). A 3D microarray was fabricated on a polymeric substrate using a nanoimprinting technique. An Au layer was deposited on the surface of the substrate by magnetron sputtering. In comparison with the smooth substrate, the substrate with the 3D surface showed a higher capture efficiency, (c.a. 4 times). In the next step, such a modified matrix was used for cancer cells immobilization and for specific determination of CTCs. An EIS technique was utilized for visualization the results. The presented solution could become a possible platform for the early diagnosis of cancer.

For the construction of microfluidic devices with an electrochemical detection system, frequently, external apparatuses have to be used, such as pumps, heaters or valves. However, in the case of designing an integrated one chip, this is not desirable. Due to this, many components, such as microvalves, with miniaturization capacity, had been developed to use in LOC apparatus to work as one chip. Here, we shortly present some systems using this element. Biswas et al. introduced a new electrochemically-actuated microvalve, which does not need any external power source and fits for autonomous stand-alone microfluidic systems [131]. A single bi-metallic zinc/platinum electrode (Zn/Pt-E) was adopted as a microvalve. The Pt area is situated in a solution microchannel, and Zn/Pt region of the electrode is located in a separated control microchannel. The Pt electrode was modified with hydrophobic self-assembled monolayer (SAM) based on, among others, 1-hexanethiol, which stops flow of the solution microchannel. Microvalves were manufactured by sputter-coating a platinum layer, and subsequent lift-off, on a substrate on which the SAM were formed. Then, such a layer was aligned and bonded to a PDMS layer. The PDMS layer was formed by replica molding technique. The microvalve is opened by injection of an electrolyte into the control microchannel, which in result oxidizes the Zn layer of the Zn/Pt electrode. Such manipulation induces a negative mixed potential of the whole electrode. Causing the negative mixed potential is able to remove SAM film from the Pt electrode surface by reductive desorption, which in result opens the microvalve and allows the solution to flow (Figure 13). As presented in that work, the microvalves-based flow system can find an application for an initiation of solution flow in any arbitrary microchannel configuration, and thereby significantly simplify peripheral equipment requirements.

A different system was developed by Chen et al.—a pump-free, capillary flow-based microfluidic chip, which can control the solution flow and can be adopted in device with electrochemical detection system [132]. It consist of an electrowetting valves combined with electrochemical transduction. For manufacturing such approach, Chen et al. used photolithography to produce a SU-8 master and thermal curing to do a PDMS stamping template for nanoimprint lithography. The designed device was made with two flexible polyethylene terephthalate (PET) layers combined to form a flexible microfluidic instrument. The first layer was constructed with UV curable thiolene (TE) polymer for microchannels forming. The second system was constructed using inkjet-printed three-electrode system and electrowetting valves were located on the opposing layer. The electrodes which contained polymeric film were bonded directly to the channel containing layer to form the sealed fluidic appliance. The described device can be used to detect *Salmonella* in a liquid sample, for application e.g., in food industry.

Satoh et al. showed a system for monitoring of ammonia metabolism in hepatocytes on a chip [133]. The concentration of ammonia was measured with an electrochemical method in the microfluidic channel. The inside structure of the chip was created from substrates based on glass and polydimethylsiloxane. Basically, the chambers for hepatocytes culture, electrolyte solution, liquid-junction reference electrode and two flow channels were manufactured in the PDMS substrate. Flow channels are isolated from each other by valves. The sensing system for ammonia included an iridium oxide pH-indicator electrode with the Ag/AgCl reference electrode. A gold working electrode and a liquid-junction Ag/AgCl electrode were employed for pH monitoring in the hepatocytes culture medium by electrolysis. The solution flow in the channels occurred by capillary force and was controlled with an electrowetting-based valves. To investigate an ammonia metabolism, ammonium ions were modified into gaseous state by increasing the pH of the cell-culture medium. The gaseous ammonia was moved by an air gap in the microfluidic channel and was able to dissolve into the solution of electrolyte, where it could be detected. The developed system allowed for the monitoring of ammonia using even one drop, and could find an application in bioanalytical investigations using valuable biological compounds.

In the future, LOC devices could be able to perform analysis virtually anywhere and under field conditions. However, mostly, it is important for the field of medical diagnostics, where the availability of a rapid, simple, low-cost, in situ whole-blood assay capable of detecting a variety of selected analytes would benefit POC testing or public health applications. The dynamic development of LOC technology in recent years, and the mutual cooperation of several scientific disciplines-biology, chemistry, electronics and material engineering, enables the development and upgrading of research tools and measurements for quick and sensitive detection. The probable application of electrochemical sensing approaches are smartphones, which are used as a data collectors [134,135]. It remains a matter of time before the use of wireless instruments—allowing for the continuous monitoring of our health parameters—is ubiquitous in our daily live.

## 7. Conclusions

Functional polymer structures related to (bio)sensor applications an LOC fabrication have recently become an important area in chemistry and engineering. Polymer-based materials, such as described in this review of CPs, MIPs or composites, possess many advantages due to their charge transport properties and (bio)sensor applications and the possibility of miniature LOC devices manufacturing. Besides the rapid development in (bio)sensing platforms for LOC technology, there are still only a few solutions present in the market which allow for continuous and multiplex analysis. As presented in this review, polymeric materials also possess the ability to act as a matrix for designing biological-based sensing approaches. The described materials represent a strong group of flexible, cost-effective, and in most cases biocompatible materials. In addition, thanks to the possibility of their chemical and physical modification, they become an excellent matrix for binding biological molecules without losing their properties. Very often they also improve receiving the analytical or bioanalytical signal, which results in higher sensitivity. Connecting an LOC device with a biological active compound biosensor system improves the selectivity of such an analytical instrument, and all the above-mentioned species represent a promising group of matrixes for biological materials anchoring. The main purpose of LOC devices is for control and evaluation, e.g., of the health status of patients, monitoring of the quality of food products and medicines, and study of changes in the environment. In our opinion, LOC technology based on polymers and (bio)sensor elements would significantly influence the development of diagnostic instruments.

## Figures and Tables

**Figure 1 polymers-12-01154-f001:**
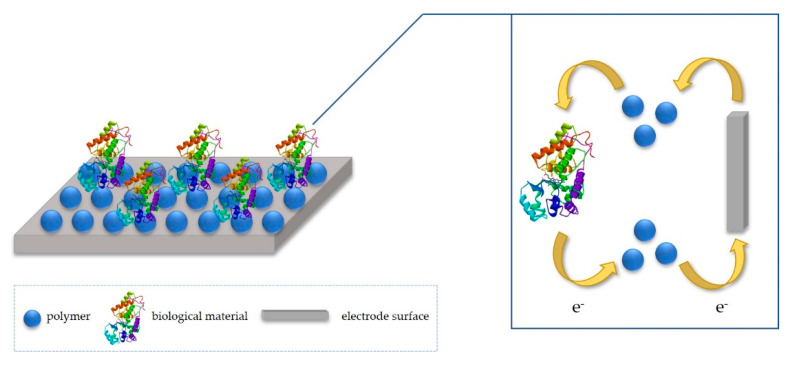
Mechanism of detection suing polymeric material.

**Figure 2 polymers-12-01154-f002:**
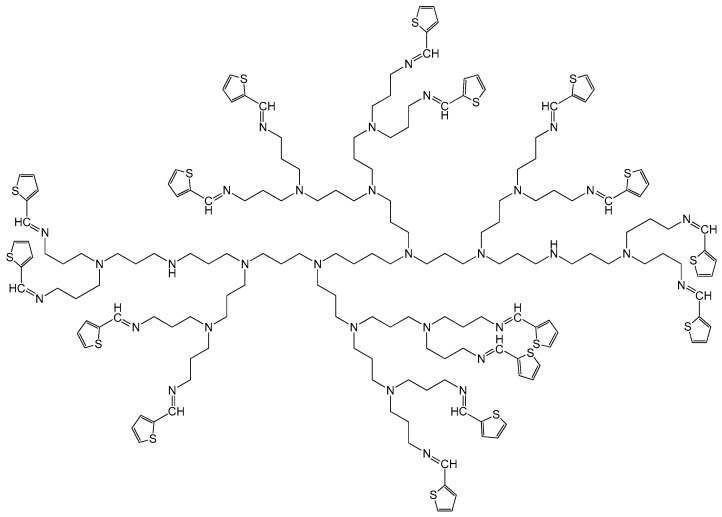
Chemical structure of 3 poly(propylenethiophenoimine) dendrimer (G3PPT).

**Figure 3 polymers-12-01154-f003:**
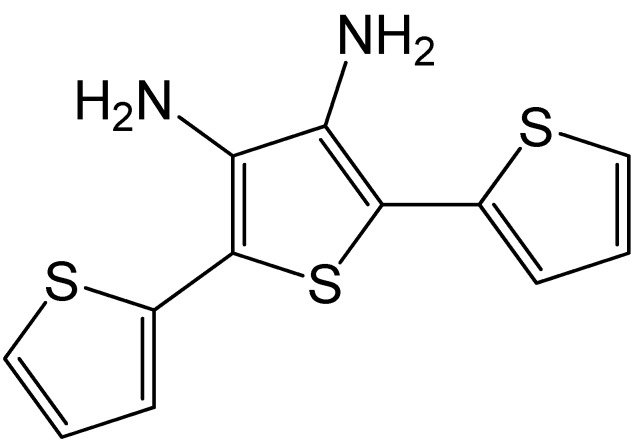
Structure of diamine functionalized terthiophene monomer [2,2′:5′,2″-terthiophene]-3′,4′-diamine (DAT).

**Figure 4 polymers-12-01154-f004:**
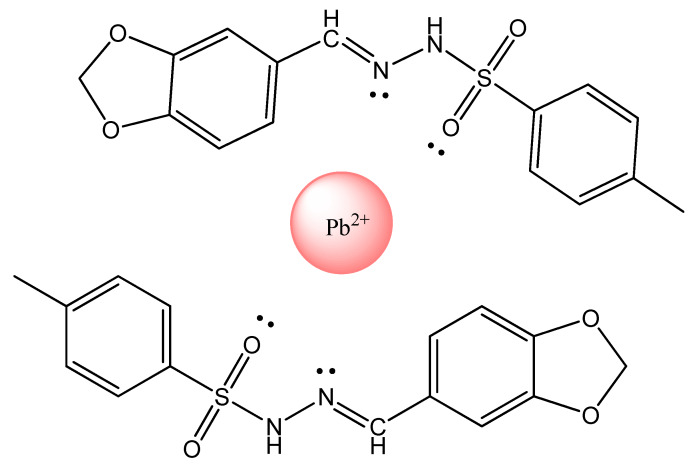
Chemical structure of (E)-N′-(benzo[d]dioxol-5-ylmethylene)-4-methyl-benzenesulfonohydrazide (BDMMBSH)- Pb^2+^ complex (proposed) based on [26].

**Figure 5 polymers-12-01154-f005:**
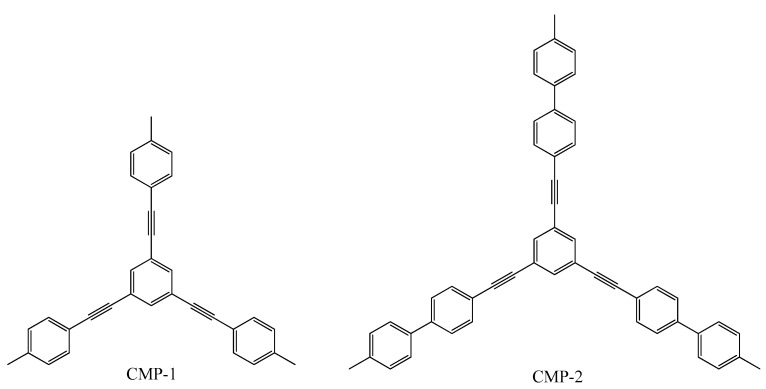
Chemical structure of 1,3,5-tris(p-tolylethynyl)benzene (CMP-1) and 1,3,5-tris((4’-methyl-[1,1’-biphenyl]-4-yl)ethynyl)benzene (CMP-2).

**Figure 6 polymers-12-01154-f006:**
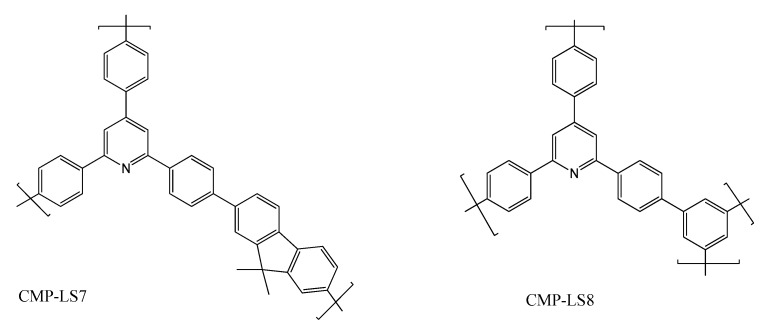
Chemical structure of CMP-LS7 and CMP-LS8.

**Figure 7 polymers-12-01154-f007:**
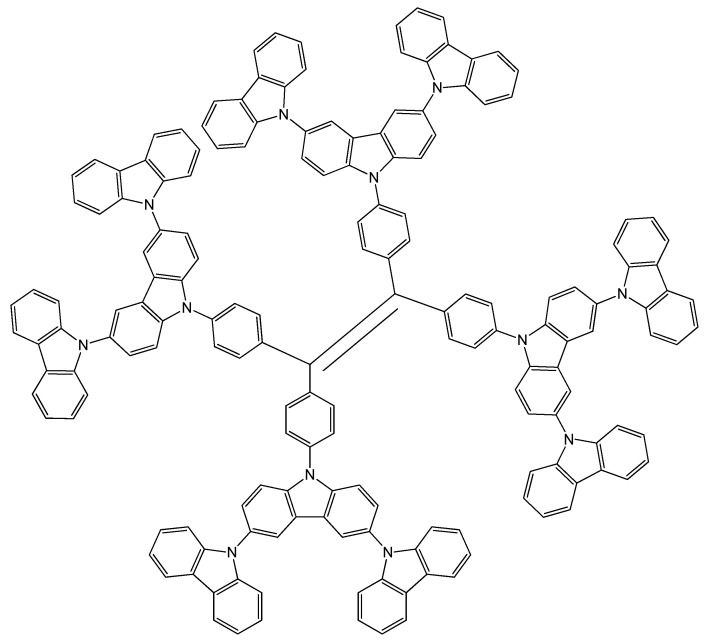
Chemical structure of dendrimer 1,1,2,2-tetrakis(4-(9’H-[9,3’:6’,9’’-*tert*-carbazol]-9’-yl)phenyl)ethene (TPETCz).

**Figure 8 polymers-12-01154-f008:**
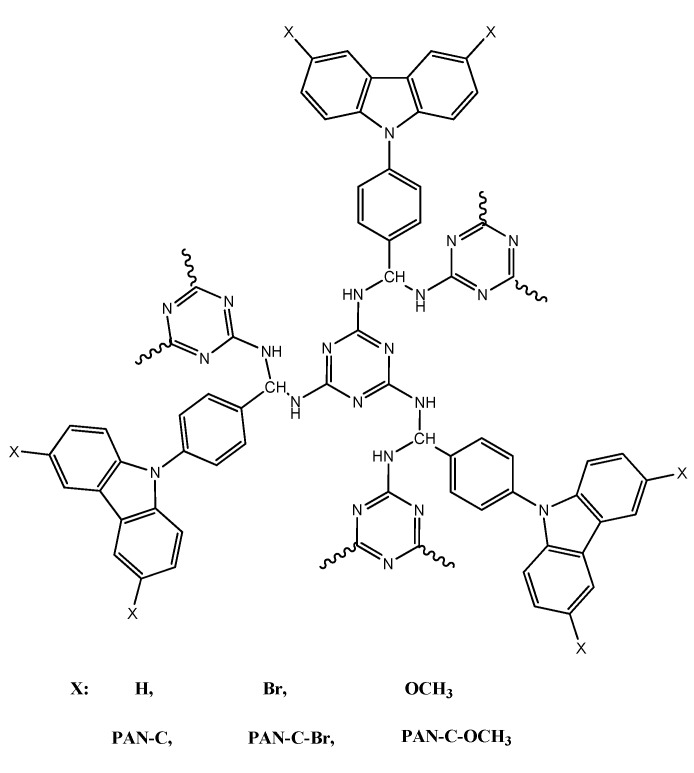
Chemical structure of carbazole-based porous polyaminals (PAN-C, PAN-C–Br, and PAN-C–OCH_3_).

**Figure 9 polymers-12-01154-f009:**
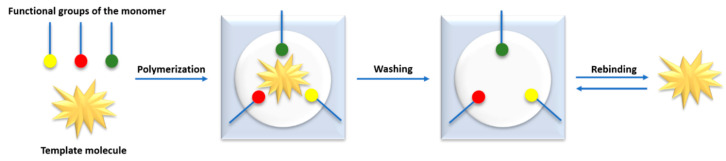
The general process of creation of molecularly imprinted polymers (MIPs).

**Figure 10 polymers-12-01154-f010:**
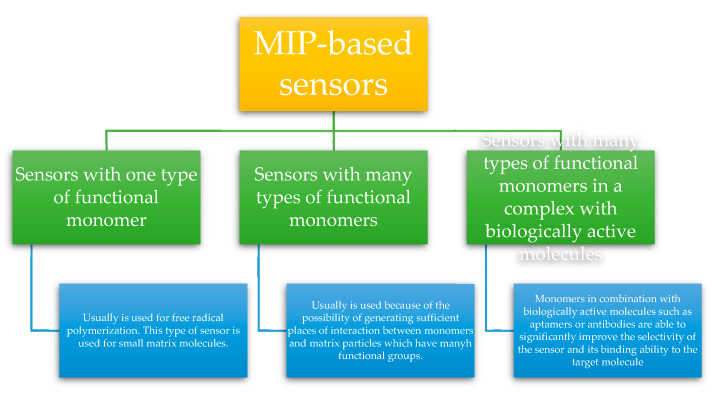
Three basic types of MIP-based sensors.

**Figure 11 polymers-12-01154-f011:**
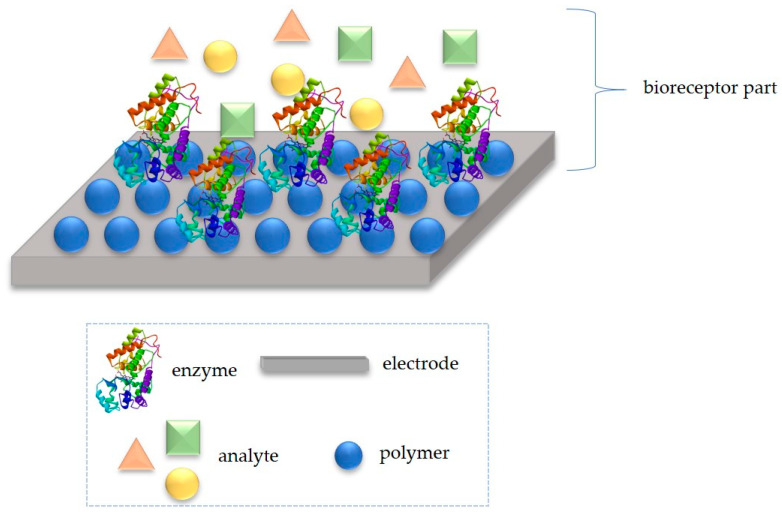
Example of bio-platform for a specific analyte recognition, which can be adopted in LOC device.

**Figure 12 polymers-12-01154-f012:**
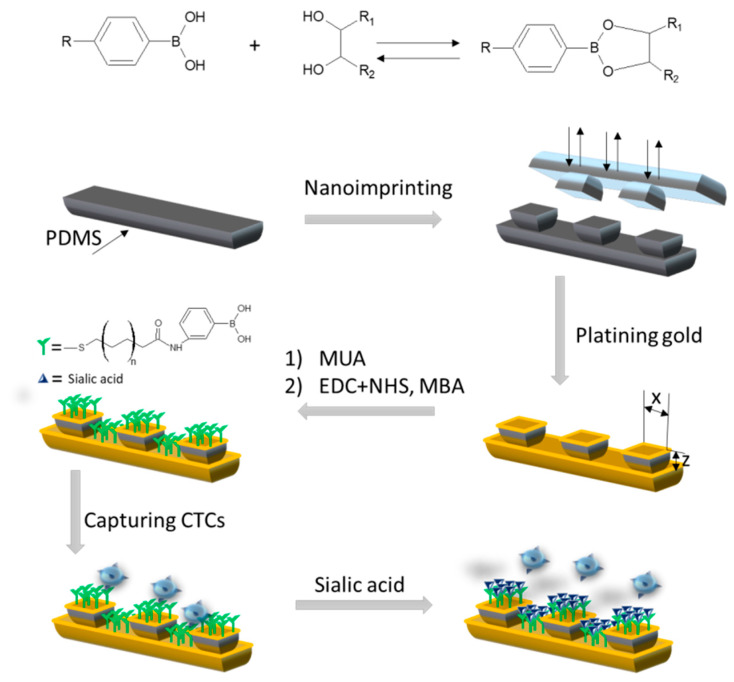
Scheme based on [129], showing the modification of gold-plated substrate, determination and circulating tumor cells (CTCs) release.

**Figure 13 polymers-12-01154-f013:**
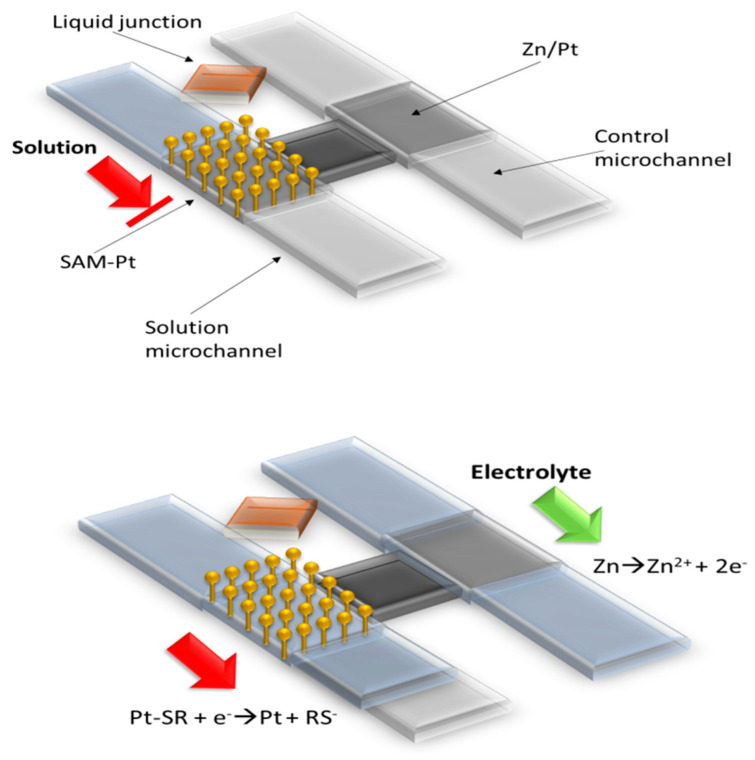
Schematic draw of electrochemically-actuated microvalve based on two electrodes (Zn/PT and modified Pt) described by Biswas et al., based on [131].

**Table 1 polymers-12-01154-t001:** The latest publication focused on the detection of various analytes by MIPs.

Template	Functional Monomer	Electrode Material	Detection Method	LOD	Ref.
17*β*-estradiol	aniline	SPCE/MIP-Fe_3_O_4_	SWV	0.02 μM	[82]
azithromycin	aniline	GCE/GO/GNU	DPV	0.1 nM	[83]
*B. cereus*	pyrrole	CPE	CV	10^2^ CFU/mL	[84]
dopamine	*o*-phenylenediamine	GCE/PPyNWs	DPV	33 nM	[85]
*E.coli*	dopamine	GCE	ECL	8 CFU/mL	[86]
epinephrine	nicotinamide	GCE/rGO	DPV	3 nM	[87]
ganciclovir	2,2′-dithiodianiline	GCE/MWCNT/AuNPs	DPAS	1.5 nM	[88]
hemoglobin	TBA	Au	CA	82 nM	[89]
HIV	*o*-phenylenediamine	ITO	ECL	0.3 fM	[90]
human serum albumin	bis(2,2-bithien-5-yl)methane	Au	DPV	0.25 pM	[91]
myoglobin	*o*-phenylenediamine	SPCE	DPV	0.006 ng/mL	[92]
PSA	dopamine	Au	MOSFET	0.1 pg/mL	[93]
serotonin	phenol	GCE/GQDs/2D-hBN	DPV	0.2 pM	[94]
s-ovoalbumin	pyrrole	GCE	DPV	2.95x10-9 mg/mL	[95]
testosterone	*o*-phenylenediamine	GCE	EIS	0.4 fM	[96]
tetracycline	pyrrole	SPCE/AuNPs	DPV	0.65 μM	[97]

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
