# Peer review of "Functional Polymers Structures for (Bio)Sensing Application—A Review"

_polymers, 2020, doi:10.3390/polym12051154_

Round 1

Reviewer 1 Report

In this manuscript, the authors Presented “Functional Polymers Structures for Lab-on-a-Chip Application – a review.  I haven’t found the consistency in this manuscript advantages. Thus, I don’t think the manuscript in its present form is suitable for publication in Polymers. Hence, major revision is needed for the acceptance of this manuscript. The details are described as follows:

  1. Introduction and abstract, needs to be free from reductant words.
  2. Introduction should be reconstructed by clearly explaining the problem statement i.e past , present and uniqueness of the work.
  3. Explain all the abbreviations at their first mention place from the Introduction section, following abbreviated term throughout the manuscript.
  4. In the Introduction section, in my opinion Authors must present a detailed comparison between fluorescent detectors and electrochemical detectors and are encouraged to add some up to date references available in recent years.

Chemical Engineering Journal 2019, 358, 101-109;

Coordination Chemistry Reviews 2019, 401, 213065;

Journal of Industrial and Engineering Chemistry 2019, 78, 315-323;

Journal of Molecular Liquids 2019, 274, 461-469

Science of the Total Environment 2018, 615, 476-485;

Science of The Total Environment 2018, 640, 174-193;

Journal of Molecular Liquids 2019, 111425;

  1. The study is similar to Literature, but in many properties the results are not as good as they are, for example, the solvent and the elucidation of mechanism etc.
  2. There is a lot of inconsistency in Figures style. Please use the same bar graph format instead.
  3. The quality of all the figures is not upto the mark therefore; all the figures must be redrawn or sketched by using better quality.

Author Response

Response to Reviewer 1 Comments

We would like to thank for Reviewer 1 detailed review, in addition we would like to inform, that every significant changes in the manuscript had been highlighted (track changes).

Point 1: Introduction and abstract, need to be free from reductant words.

Response 1: Thank you for this comment. The abstract and introduction have been modified to remove redundant words as suggested.

Point 2: Introduction should be reconstructed by clearly explaining the problem statement i.e. past, present and uniqueness of the work.

Response 2: An introduction part has been modified, as suggested.

Point 3: Explain all the abbreviations at their first mention place from the Introduction section, following abbreviated term throughout the manuscript.

Response 3: All abbreviations have been explained at their first mention place. The rest of the manuscript has been standardized to previously introduced abbreviations.

Point 4: In the Introduction section, in my opinion Authors must present a detailed comparison between fluorescent detectors and electrochemical detectors and are encouraged to add some up to date references available in recent years.

Response 4: Thank you for this comment. The comparison between these two detectors has been added, as suggested with proposed references.

Point 5: The study is similar to Literature, but in many properties the results are not as good as they are, for example, the solvent and the elucidation of mechanism etc.

Response 5: All described studies, as an examples for using conducting materials in sensors has been refined with all important information, as suggested.

Point 6: There is a lot of inconsistency in Figures style. Please use the same bar graph format instead.

Response 6: Due to the Reviewer suggestion, the figures had been modified.

Point 7: The quality of all the figures is not up to the mark therefore; all the figures must be redrawn or sketched by using better quality.

Response 7: Figures in the manuscript had been checked once again, and, where necessary, improved.

Reviewer 2 Report

The authors review on the utilization of “Functional Polymers Structures for Lab-on-a-Chip Application”.

The review in general is well-structured, concise and containing novel articles relevant to the field.

Some minor comments and corrections below:

-Line 14: materials’

-Line 18: please change the order of the keywords; for this review it is not the most important first keyword to appear i.e. “composites”.

-Line 22: please change with the following phrase: “An increasing interest in the development...”

-Line 35: a reference in missing.

-Line 90: please change “paper” with “review”

-Throughout the whole manuscript, for the different sensor, the authors describe nicely the sensors’ material, novelty and species of detection, the LOD, etc. However it is important the authors to include 2-3 lines for all related works regarding the sensor device characteristics i.e. two electrode resistive sensor or three electrode electrochemical capacitive (or DVP measurement sensor, etc.), or electrochemical transistor sensor, etc. as well as possible the fabrication method i.e. spin coating for the sensing layer, screen printing of the underlying IDE electrode, or metallic contact deposition with e-beam evaporation especially in cases of electrochemical transistors, etc. Moreover, it should be defined the type of sensor i.e. electrical sensor or optical sensor and the medium that the sensor works i.e. liquid or gas sensor (in cases within the manuscript that is not highlighted).

-For instance, please follow the very nice way (explanatory and educational) as for the sensor system it is described in line 273-285.

-Line 545: please improve the figure quality and correct the figure number in the caption.

-Line 615: is the Au layer deposited by electroplating or by e-beam evaporation. Please these very basic device fabrication information should be given precisely for all sensor systems presented/ discussed in this review article.

In terms of originality, scientific quality, relevance & contribution to the field and presentation, this review article is of good level. However, the authors should include some details for all the sensor systems presented in order to be more attractive and educational for the reader considering that the community that will read this review could be from various fields and different backgrounds i.e. chemistry, physics, polymer science and synthesis, chemistry of materials, electrochemistry, device fabrication, materials science, biology and biochemistry, electrical and electronic engineering, etc.

Furthermore, the discussion should be improved in the points that have been indicated above.

The review article is sufficiently novel to warrant its publication, however, after including and considering all the changes proposed.

Author Response

Response to Reviewer 2 Comments

We would like to thank for Reviewer 2 detailed review, in addition we would like to inform, that every significant changes in the manuscript had been highlighted (track changes).

Point 1: Line 14: materials’

Response 1: The word has been corrected.

Point 2: Line 18: please change the order of the keywords; for this review it is not the most important first keyword to appear i.e. “composites”.

Response 2: The order of the keywords has been changed.

Point 3: Line 22: please change with the following phrase: “An increasing interest in the development...”

Response 3: The phrase has been changed as suggested.

Point 4: Line 35: a reference in missing.

Response 4: The introduction has been modified. All references have been included. The numbering of subsequent references has been updated.

Point 5: Line 90: please change “paper” with “review”.

Response 5: The word “paper” has been changed with “review”.

Point 6: Throughout the whole manuscript, for the different sensor, the authors describe nicely the sensors’ material, novelty and species of detection, the LOD, etc. However it is important the authors to include 2-3 lines for all related works regarding the sensor device characteristics i.e. two electrode resistive sensor or three electrode electrochemical capacitive (or DVP measurement sensor, etc.), or electrochemical transistor sensor, etc. as well as possible the fabrication method i.e. spin coating for the sensing layer, screen printing of the underlying IDE electrode, or metallic contact deposition with e-beam evaporation especially in cases of electrochemical transistors, etc. Moreover, it should be defined the type of sensor i.e. electrical sensor or optical sensor and the medium that the sensor works i.e. liquid or gas sensor (in cases within the manuscript that is not highlighted).

Response 6: Exemplary sensor descriptions have been enriched with additional information in accordance with Reviewer suggestion.

Point 7: Line 545: please improve the figure quality and correct the figure number in the caption.

Response 7: The figure quality and captions has been improved.

Point 8: Line 615: is the Au layer deposited by electroplating or by e-beam evaporation. Please these very basic device fabrication information should be given precisely for all sensor systems presented/ discussed in this review article.

Response 8: The description of sensors had been improved with necessary information.

Point 9: The discussion should be improved in the points that have been indicated above.

Response 9: The conclusions has been matched to the review topic.

Round 2

Reviewer 1 Report

Accept

Reviewer 2 Report

The authors have made all changes and the current version can be accepted for publication.